# LEARNING DISCRIMINATIVE REPRESENTATIONS FOR CHROMOSOME CLASSIFICATION WITH SMALL DATASETS

## ABSTRACT

Chromosome classification is crucial for karyotype analysis in cytogenetics. Karyotype analysis is a fundamental approach for clinical cytogeneticists to identify numerical and structural chromosomal abnormalities. However, classifying chromosomes accurately and robustly in clinical application is still challenging due to: 1) rich deformations of chromosome shape, 2) similarity of chromosomes, and 3) imbalanced and insufficient labelled dataset. This paper proposes a novel pipeline for the automatic classification of chromosomes. Unlike existing methods, our approach is primarily based on learning meaningful data representations rather than only finding classification features in given samples. The proposed pipeline comprises three stages: The first stage extracts meaningful visual features of chromosomes by utilizing ResNet with triplet loss. The second stage optimizes features from stage one to obtain a linear discriminative representation via maximal coding rate reduction. It ensures the clusters representing different chromosome types are far away from each other while embeddings of the same type are close to each other in the cluster. The third stage is to identify chromosomes. Based on the meaningful feature representation learned in the previous stage, traditional machine learning algorithms such as SVM are adequate for the classification task. Evaluation results on a publicly available dataset show that our method achieves 97.22% accuracy and is better than state-of-the-art methods.

## 1 INTRODUCTION

Human chromosome classification is crucial for karyotype analysis in cytogenetics. Karyotype analysis is a fundamental approach for clinical cytogeneticists to identify numerical and structural chromosomal abnormalities, such as Turner syndrome, Chronic myelogenous leukaemia, Edwards syndrome, and Down syndrome (Stebbins & Ledyard., 1950; Sharma et al., 2017). In clinical practice, karyotyping requires the preparation of a complete set of micro-photographed metaphase chromosomes in the cells, or more precisely, a karyogram (Figure 1). To do so, the cytogeneticists need to classify and sort these chromosomes into 23 pairs of chromosomes, including 22 pairs of autosomes and a pair of sex chromosomes (X and Y chromosomes in male cells and double X in female cells) (Jindal et al., 2017).

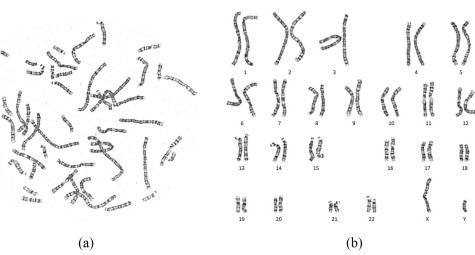

Figure 1: (a) A G-stained microscopic image of male chromosomes for one case. (b) The karyogram of (a).

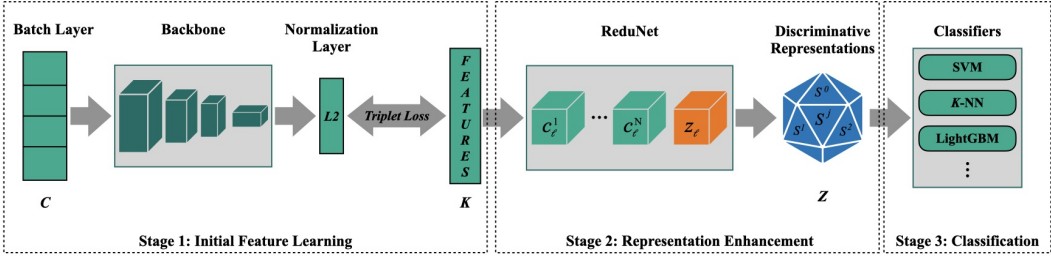

Figure 2: The flow diagram of our proposed pipeline: Chromosome images $C$ are processed using the Backbone network (ResNet) with Triplet Loss for initial feature Learning. Then the learned feature embeddings $K$ are fed to the ReduNet for discriminative representation learning. Finally, the enhanced feature representations $Z$ are used for chromosome classification by the classifier.

Chromosomes are highly coiled and condensed in metaphase, making karyotyping challenging, complicated, and laborious. Even for experienced cytogeneticists, considerable time and manual effort are indispensable in classifying and sorting various types of chromosomes to produce a karyogram. In order to reduce the burden of karyotyping, many researchers (Piper & Granum., 1989; Errington et al., 1993; Mashadi et al., 2007; Sharma et al., 2018; Lin et al., 2020) have been dedicated to auto-karyotyping using computer powers for decades. However, classifying chromosomes accurately and robustly in clinical application is still challenging. These challenges mainly stem from three aspects( Jindal et al., 2017; Lin et al., 2020): 1) rich deformations of chromosome shape, 2) similarity of chromosomes, and 3) imbalanced and insufficient labelled dataset.

To tackle the above challenges, in this paper, we propose a novel pipeline for the automatic classification of chromosomes (Figure 2). However, unlike existing methods, our approach is primarily based on learning meaningful data representations rather than only finding classification features in given samples. Specifically, our proposed pipeline comprises three stages: The first stage extracts meaningful visual features of chromosomes by utilizing ResNet (He et al., 2016) with triplet loss (Weinberger et al., 2009; Schroff et al., 2015). The second stage optimizes features from stage one to obtain a linear discriminative representation via maximal coding rate reduction (Chan et al., 2021). It ensures the clusters representing different chromosome types are far away from each other while vectors of the same type are close to each other in the cluster. The third stage is to identify chromosomes. Based on the meaningful feature representation learned in the previous stages, traditional machine learning algorithms such as the Support Vector Machine (SVM) (Boser et al., 1992) can easily be plugged into our pipeline as classifiers.

One public dataset is used to validate the performance and generalizability of our approach. Extensive experiments on the datasets corroborate that our proposed method achieved better performance than state-of-the-art methods. Our contributions can be summarised as follows:

- We propose a representation learning-oriented approach for chromosome classification. Based on the enhanced feature representation, simple and efficient machine learning algorithms are adequate for the classification task.

- Inspired by FaceNet's (Schroff et al., 2015) capability of distinguishing similar faces and ReduNet's competence in representation structure optimization, we propose combining them for more meaningful and effective feature learning with a small dataset.

- We evaluate the proposed approach on a public dataset. It demonstrates its superior performance compared with the state-of-the-art method.

## 2 RELATED WORK

In the early years, hand-crafted and geometrical features (e.g., a chromosome's axis, centromere position, banding pattern features, and length) and complex pre-processings like straightening the chromosomes are primarily required with traditional machine learning algorithms for automatic chromosome classification (Ledley & Lubs., 1980; Egmont-Petersen et al., 2002; Javan-Roshtkhari

et al., 2007; Jahani et al., 2012). Recently, with the significant achievement of deep learning (LeCun et al., 2015), researchers tended to employ deep learning-based methods in chromosome classification. In research (Jindal et al., 2017), Jindal et al. proposed to use the Siamese Networks (Koch et al., 2015), a deep learning-based approach, for chromosome classification. First, they straighten all chromosomes and then use the Siamese Networks to push the embeddings of similar samples closer. Finally, the classification is carried out via an MLP-based module on top of the embeddings given by the trained Siamese Networks. The experimental results yield 84.6% classification accuracy on their private dataset. Sharma et al. (Sharma et al., 2017) proposed a convolutional neural network (CNN) based method for classification. Curved chromosomes were straightened by cropping and stitching and then normalized by the length calculated according to centromere position. Based on such pre-processing, their method achieved 86.7% classification accuracy. Monika et al. (Sharma et al., 2018) proposed an attention-based (Vaswani et al., 2017) pattern-sequence learning method for chromosome classification. By combining the residual convolutional neural network with the recurrent attention neural network, their method exploits the local features of chromosome bands and the property of chromosome band sequences. The evaluation results showed that their method achieved 91.94% classification accuracy. Yulei et al. (Qin et al., 2019) proposed a global-local combined approach for chromosome classification composed of global and local-scale networks. The global network is used to obtain features and find local regions. The found regions are fed into the local network using a varifocal mechanism for local feature extraction. They further promote high-level feature extraction through residual and multi-task learning strategies. The authors claimed that this method yielded 99.2% classification accuracy on their large dataset (i.e. 1909 karyotyping cases, 87814 G-band chromosome images). Lin et al. (Lin et al., 2020) introduced an automatic chromosome classification approach based on an optimized version of Inception (Szegedy et al., 2017) and a simple but effective augmentation method called CDA for performance improvement. Their data augmentation algorithm not only can enlarge the training dataset and improve the accuracy and robustness of the classifier but also can eliminate the directional features of chromosomes, which allow the model to classify chromosomes without special pre-process (e.g., rotating, straightening). The experimental results showed that their method achieved 95.98% classification accuracy on the clinical G-band chromosome dataset.

Although the above methods seem to solve the chromosome classification problem to some extent, there are still some limitations to these methods. First, to tackle the deformation problem, the pre-processing operation, such as straightening, is commonly required in many previous approaches, which brings an extra step to the clinical application and increases the complexity of the classification task. Second, most existing methods are classification task-oriented, which do not learn patterns of chromosomes but focus on finding features that can distinguish limited samples. Such classification task-oriented design leads to poor generalizability when tackling chromosome data rich in shape deformations and are very similar. Last, for most of these methods, an extensive dataset is desired. However, sample labelling is labour-intensive and expensive, which restricts the performance of these methods to a relative level according to their dataset size.

## 3 METHOD

The proposed method for chromosome classification consists of three stages (Figur 2): the first stage is the initial extraction and embedding of features using a deep convolutional neural network with triplet loss. The second stage is to enhance the effectiveness and structure of the representations of feature embeddings via the ReduNet. The third stage is classifying the chromosome types based on enhanced data representations.

### 3.1 STAGE 1: INITIAL FEATURE LEARNING

It is common for chromosomes to look similar (Figure 1). Additionally, some types of chromosomes are very hard to distinguish in length and banding. Together with the effect of highly coiled and condensed chromosomes and low-resolution micro-photograph, it becomes even tougher to identify chromosomes. Inspired by FaceNet's (Schroff et al., 2015) capability of distinguishing similar faces via triplet loss, we propose combining ResNet with triplet loss to extract more meaningful and effective features of chromosomes.

The architecture of the initial feature learning comprises three main elements: one batch input layer, one backbone network (i.e. ResNet), and one normalization layer producing the feature embeddings. The batch layer prepares the target sample along with positive and negative samples. The backbone network learns features from these triplet samples through the triplet loss function at the network's end. Moreover, the learned features are embedded as vectors at the normalization layer as the output of this stage.

To learn the ideal features, we aim to embed an image $c$ into a feature space $\mathbb{R}^d$ through the backbone network $f$ with parameters $\theta$, such that the distance between all chromosomes of the same class is short, whereas the distance between a pair of chromosome images from different classes is long. The learned feature embedding is represented as $k = f_\theta(c) \in \mathbb{R}^d$, it embeds an image $c$ into a $d$-dimensional Euclidean space. Additionally, we constrain this embedding to live on the $d$-dimensional hypersphere, i.e. $\| k \|_2 = 1$. Here we want to ensure that an image $c_i^t$(target chromosome) of a particular chromosome is closer to all other images $c_i^p$(positive) of the same class than it is to any image $c_i^n$(negative) of any other classes. Formally, we want:

$$\| k_i^t - k_i^p \|_2^2 + \alpha < \| k_i^t - k_i^n \|_2^2 . \tag{1}$$

$$\forall (k_i^t = f_\theta(c_i^t), k_i^p = f_\theta(c_i^p), k_i^n = f_\theta(c_i^n)) \in \tau. \tag{2}$$

Where $\alpha$ is a margin that enforces the distance between positive and negative pairs, $\tau$ is the set of all possible triplets in the training set of size U. Then, we can have the loss function as:

$$L = \sum_i^U \left[ \| f(c_i^a) - f(c_i^p) \|_2^2 - \| f(c_i^a) - f(c_i^n) \|_2^2 + \alpha \right] . \tag{3}$$

In order to ensure fast convergence and learn a meaningful embedding, we introduce the batch input layer to ensure that a minimal number of exemplars of any specific class are present in each mini-batch. To this end, we can train the ResNet with the triplet loss via Stochastic Gradient Descent (SGD) (Wilson et al., 2003).

## 3.2 STAGE 2: REPRESENTATION ENHANCEMENT

It would be enough to learn good features through stage one if the data were not rich in shape deformation and there were sufficient samples. However, chromosomes have non-rigid intrinsic nature and are highly coiled and condensed in metaphase. Hence, it is critical to represent features in low-dimensional structures invariant to such deformations, which are known to have sophisticated geometric and topological structures and can be challenging to learn precisely, even with rigorously designed CNNs (Cohen et al., 2019; Chan et al., 2021). Therefore, we employ ReduNet for a discriminative linear representation after the initial feature learning to overcome this issue.

### 3.2.1 DISCRIMINATIVE REPRESENTATION LEARNING

The objective of the representation enhancement is to obtain intrinsic low-dimensional structures from the high-dimensional data, and the learned structures/representation should meet the requirements (Chan et al., 2021) : 1) *Compressible for same class*: Features of samples from the same class/cluster should be relatively correlated in the sense that they belong to a particular low-dimensional linear subspace and are close to each other, 2) *Discriminative for deferent classes*: Features of samples from different classes/clusters should be highly uncorrelated and belong to different low-dimensional linear subspaces, and 3) *Diverse Representation*: The dimension of features for each class/cluster should be as large as possible as long as they stay uncorrelated from the other classes.

Formally, we want to learn a representation $z = g_{\theta'}(k)$ that maps each of the submanifolds $M^j \subset \mathbb{R}^D$ to a linear subspace $S^j \subset \mathbb{R}^d$. Given initially learned chromosome feature samples $K = [k_1, k_2, ..., k_m]$, we can have a finite number of samples as learned representations $z_j = g_{\theta'}(k) \in \mathbb{R}^d, i = 1, 2, ..., m$. To satisfy the above requirements, measuring the distribution compactness of a random representation $z$ or its finite samples $Z = [z_1, z_2, ..., z_m]$ is fundamental. First, we estimate the number of binary bits needed to encode the learned representation $Z$ up to a precision $\epsilon$, which can be given by:

$$\mathcal{L}(Z, \epsilon) \doteq \left( \frac{m + d}{2} \right) \log \det \left( I + \frac{d}{m\epsilon^2} ZZ^T \right) . \tag{4}$$

Where the function $\det \log(\cdot)$ is very effective in clustering or classification of mixed data with guaranteed convergence (Fazel et al., 2003; Ma et al., 2007; Kang et al., 2015). By computing the number of bits needed to quantize the SVD of $Z$ subject to the precision or by packing $\epsilon$-balls into the space spanned by $Z$ as a Gaussian source, we can derive (4). Please refer to (Chan et al., 2021; Fazel et al., 2003) for more technical details and proofs.

Then, the compactness of the learned features can be measured in terms of the average coding length per sample, a.k.a. the coding rate subject to the distortion $\epsilon$ :

$$R\left(Z, \epsilon\right) \doteq \frac{1}{2} \log \det \left(I + \frac{d}{m\epsilon^2} ZZ^T\right). \tag{5}$$

### 3.2.2 OPTIMIZATION

To make the learned representations discriminative, features of different classes/clusters should be maximally incoherent to each other. Hence we can stretch $Z$ to allow the coding rate of the whole set to be as large as possible. Meanwhile, for each class/cluster, the learned representations should be highly correlated and coherent. Therefore, we need to compress each class/cluster's space (or subspace) to a minimal volume to make the subspace coding rate as small as possible.

In general, the representation $Z$ is a mixed data comprising multiple low-dimensional subspaces of different classes. To evaluate the rate-distortion of such mixed data more accurately, we partition $Z$ into multiple subsets: $Z = Z^1 \cup Z^2... \cup Z^N$ with each $Z^j$ containing samples in one low-dimensional subspace. Let $\wedge = \{\wedge_j \in \mathbb{R}^{m \times m}\}_{j=1}^N$ be a set of diagonal matrices with diagonal entries encoding the associates of the $m$ samples in the $N$ classes. Precisely, the diagonal entry $\wedge^j(i,i)$ of $\wedge^j$ reveals the probability of sample $i$ belonging to the subset $j$. Thus, we can have the simplex: $\Phi \doteq \{\wedge|\wedge^j \geq 0, \wedge^1 + ... + \wedge^j = I\}$. With respect to this partition, the average number of bits per sample (the coding rate) can be obtained from (Ma et al., 2007):

$$R_c(Z, \epsilon|\wedge) \doteq \sum_{j=1}^N \frac{tr(\wedge^j)}{2m} \log \det(I + \frac{d}{tr(\wedge^j)\epsilon^2} Z \wedge^j Z^T). \tag{6}$$

Therefore, given a partition $\wedge$ of $Z$ to make a significant difference between the coding rate of the whole and all the subsets can be achieved by maximizing the coding rate reduction:

$$\max_{\theta' \wedge} \Delta R(Z, \wedge, \epsilon) = R(Z, \epsilon) - R_c(Z, \epsilon|\wedge),$$
$$\text{s.t. } \| Z^j(\theta') \|_F^2 = m_j, \wedge \in \Phi \tag{7}$$

Where the Frobenius norm normalizes the learned features to scale with the number of features in $Z^j \in \mathbb{R}^{d \times m_j}$ : $\| Z^j(\theta') \|_F^2 = m_j$. As a result, the $\Delta R$ can be maximized as a function of $Z \subset \mathbb{S}^{d-1}$ by using gradient ascent in a deep network (i.e. ReduNet) with a purely forward propagation fashion:

$$Z_{\ell+1} \propto Z_\ell + \eta \cdot \frac{\partial \Delta R}{\partial \Delta Z}\Big|_{Z_\ell} \quad \text{s.t.} \quad Z_{\ell+1} \subset \mathbb{S}^{d-1}. \tag{8}$$

Where $\ell$ indicates the layer, $Z_\ell = \left[z_\ell^1, ..., z_\ell^m\right]$, $\eta$ is the step size larger than 0, and the iteration starts with the given data $Z_1 = K$, The gradient $\frac{\partial \Delta R}{\partial \Delta Z}$ entails two derivatives of expanding and compressing in $\Delta R(Z)$, respectively:

$$E_\ell = \alpha(I + \alpha Z_\ell Z_\ell^T)^{-1}, \tag{9}$$

$$C_\ell^j = \alpha_j(I + \alpha_j Z_\ell \wedge^j Z_\ell^T)^{-1}. \tag{10}$$

$$\frac{\partial \Delta R}{\partial Z}\Big|_{z_\ell} = \underbrace{E_\ell}_{Expansion} - \sum_{j=1}^k \gamma_j \underbrace{C_\ell^j}_{Compression} Z_\ell \wedge^j . \tag{11}$$

- $E_\ell$ : Expansion of the features sapce at the $\ell^{th}$ layer

- $C_\ell^j$ : Compression of the $j^{th}$ class features at the $\ell^{th}$ layer.

Finally, to find the optimal $g_{\theta'}(k)$, we create a small increment transform at the $\ell^{th}$ layer's feature $z_\ell$ to carry out the above gradient ascent optimization, which can be approximated by a nonlinear activation function $\sigma(\cdot)$ in the neural networks as:

$$Z_{\ell+1} \propto Z_\ell + \eta \cdot E_\ell z_\ell - \eta \cdot \sigma\left(\left[C_\ell^1 z_\ell, ..., C_\ell^N z_\ell\right]\right). \tag{12}$$

In ReduNet, the rectified linear unit operation, $ReLu$, is used in this approximation as:

$$\sigma(Z_\ell) \propto Z_\ell - \sum_{j=1}^{N} ReLu(P_\ell^j Z_\ell). \tag{13}$$

Where $P_\ell^j = (C_\ell^j)^\perp$ is the projection onto the $\ell^{th}$ class so that the optimization can be performed through its layers, as shown in Figure 3.

To this end, the initial features are transformed into an optimized distribution by ReduNet through this gradient ascent operation, where the representation space for the whole set is expanded as large as possible, and representations of the same classes are compressed as small as possible.

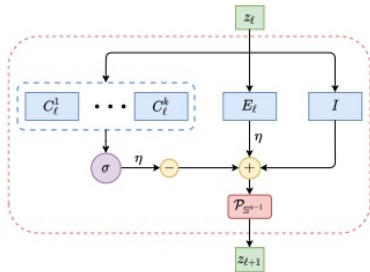

Figure 3: A layer of ReduNet derived from one iteration of gradient ascent for the rate reduction optimization.

### 3.3 STAGE 3: CLASSIFICATION

Through the previous two stages, we map the chromosome images into a linear discriminative representation. As the representation is ready, we can classify chromosomes with conventional pattern recognition algorithms.

In our proposed method, an SVM is used as the classifier. SVM uses an optimum linear separating hyperplane to separate two data sets in feature space (Mashadi et al., 2007). This optimum hyperplane isproduced by maximizing the minimum margin between the two sets. Therefore the resulting hyperplane will only be depended on border training patterns called support vectors.

Specifically, the kernel function has an essential effect on the performance of SVM. In this paper, we use one of the popular kernel functions, the Gaussian kernel.

## 4 EXPERIMENTS AND RESULTS

### 4.1 DATASET

To evaluate the performance of our proposed method, we use the publicly available dataset reported in the study (Lin et al., 2020). There are only 65 cases total (32 male and 33 female karyotypes). Usually, one case comprises 46 individual chromosomes, which means 2990 total chromosome samples for 24 classes. Specifically, for 22 autosomes (labelled as 0 to 21), there are 130 samples per class, while for sex chromosomes, there are 98 ($33 \times 2 + 32$) X samples labelled as 22 and only 32 Y samples labelled as 23. We split the dataset into training and testing sets using the ratio of 90:10.

### 4.2 Implementation Details

We first uniform the input image size as 224×224 pixels and then normalize all chromosome images $C$ as follows: $c'_j = c_j/255, \quad i = 1, 2, ..., N$. Because the amount of the Y chromosome is too small, we replicated it four times in our experiment. The images are randomly cropped and flipped for data augmentation during the training process. Random cropping is performed to simulate the effects of crossing and covering among chromosomes, and random flipping augments the position of chromosomes. In stage one, the ResNet-50 is used as the backbone network. All modules in this paper are trained using the SGD optimizer. The initial learning rate is set to 0.001, which reduces by 50 per cent every ten epochs, in a total of 50 epochs. All proposed modules are implemented in Python with the PyTorch framework. All experiments are conducted under a Ubuntu OS workstation with one single NVIDIA RTX 3090 GPU.

### 4.3 Evaluation metrics

Four metrics are used to evaluate the performance of our method: the accuracy of all the testing chromosome images ($Acc$), the average precision over classes of all the testing images ($P$), the average recall over classes of all the testing images ($Recall$), and the average F1 score over classes of all the testing images ($F_1$) as:

$$Acc = \frac{1}{N} \sum_{j=1}^{N} \frac{TP_j + TN_j}{TP_j + TN_j + FP_j + FN_j}, \tag{14}$$

$$Recall = \frac{1}{N} \sum_{j=1}^{N} \frac{TP_j}{TP_j + FN_j}, \tag{15}$$

$$Precision = \frac{1}{N} \sum_{j=1}^{N} \frac{TP_j}{TP_j + FP_j}, \tag{16}$$

$$F_1 = \frac{1}{N} \sum_{j=1}^{N} \frac{2 \cdot Precision_j \cdot Recall_j}{Precision_j + Recall_j}, \tag{17}$$

where $TP$ is the number of true positive instances, $TN$ is the number of true negative instances, $FP$ is the number of false positive instances, and $FN$ is the number of false negative instances, $N$=24 represents the 24 classes of chromosomes.

### 4.4 Results

This subsection presents the experimental results in two parts. First, we provide the classification results of our proposed method compared with state-of-the-art methods. Then, we provide the performance analysis details.

#### 4.4.1 Comparison with the State-of-the-Art

The general experimental results of our proposed method and several other methods are shown in Table 1. We use three basic CNN networks, AlexNet (Krizhevsky et al., 2012), VGG-16 (Simonyan & Zisserman, 2014), and ResNet- 50 (He et al., 2016), as the baseline methods and three advanced methods reported in (Lin et al., 2020) to compare with. All experiments were conducted using the same dataset.

According to the experimental results, our proposed approach achieves 97.22% classification accuracy, which is better than other methods. It further improves the classification performance by 13.01 percent of the Acc. metric compared to the ResNet-50 baseline, 9.59 percent to SiameseNet (Lin et al., 2020), and 1.24 percent to the CIR-Net. To the best of our knowledge, CIR-Net is the state-of-the-art method on this public dataset.

Moreover, the receiver operating characteristic (ROC) curves and the corresponding area under the curve (AUC) are popular measurements used for performance analysis. Although they are general

Table 1: Performances of our method compared with state-of-the-art methods

| Method | $Acc(\%)$ | $F_1(\%)$ | $P(\%)$ | $Recall(\%)$ |
|---|---|---|---|---|
| AlexNet(Krizhevsky et al., 2012) | 83.86 | 83.52 | 84.83 | 83.56 |
| VGG-16(Simonyan & Zisserman, 2014) | 91.22 | 91.22 | 92.21 | 91.20 |
| ResNet-50(He et al., 2016) | 91.92 | 91.72 | 91.78 | 92.44 |
| Vanilla-CNN(Lin et al., 2020) | 86.44 | 87.00 | 88.00 | 86.00 |
| SiameseNet(Lin et al., 2020) | 87.63 | 87.00 | 88.00 | 97.00 |
| CIR-Net(Lin et al., 2020) | 95.98 | 96.00 | 96.00 | 96.00 |
| ResNet+ReduNet (ours) | 92.63 | 92.39 | 97.54 | 92.94 |
| ResNet+TripLet-Loss (ours) | 96.68 | 96.43 | 96.80 | 96.80 |
| ResNet+TripLet-Loss+ReduNet (ours) | **97.22** | **97.18** | **97.43** | **97.22** |

evaluation metrics for binary classification problems, we converted them in order to support multi-classification problems: A one-to-all scheme is first used for the ROC analysis of each class, and then we averaged the ROC curves for all categories and calculated the AUC for each method. The higher AUC values, the better performance (Figure 4). It can be observed that our proposed method outperforms other methods with the least number of false positive predictions and the highest true positive rate. Our proposed method achieves the highest AUC for the classification task.

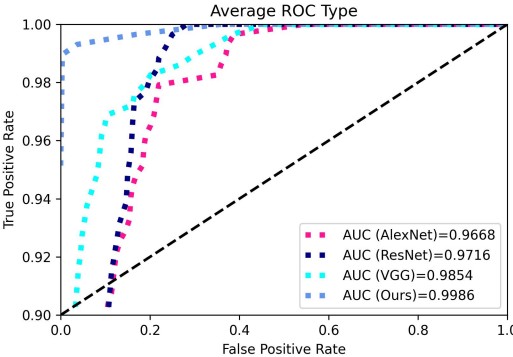

Figure 4: Comparsions on ROC curves and AUC values of our proposed method and other CNN models.

### 4.4.2 PERFORMANCE ANALYSIS

The experimental results (Table 1) show that the classification performance of conventional CNNs are all suboptimal, which suggests that the high similarity of chromosomes is a big issue for normal CNNs. Even though the ResNet-50 is 8.06 percent more accurate than AlexNet and achieves the best classification accuracy among conventional CNNs, it is still 4.76 percent less accurate than ResNet with triplet loss and 5.3 percent less accurate than ResNet with triplet loss + ReduNet. When we only apply the ReduNet to ResNet-50, the classification accuracy increases by 0.71 percent, while the accuracy rises from 91.92% to 96.68% when only adding triplet loss to ResNet-50. We argue that this is because the ResNet-50 cannot learn effective chromosome features in the first place, leading the ReduNet can hardly do more with these features. Therefore, we first apply the triplet loss to ResNet-50 to help learn more meaningful features, then use ReduNet to enhance the learned features. As the results show, the classification accuracy of our proposed method further improves by 0.54 percent when applying the ReduNet to ResNet-50 with triplet loss.

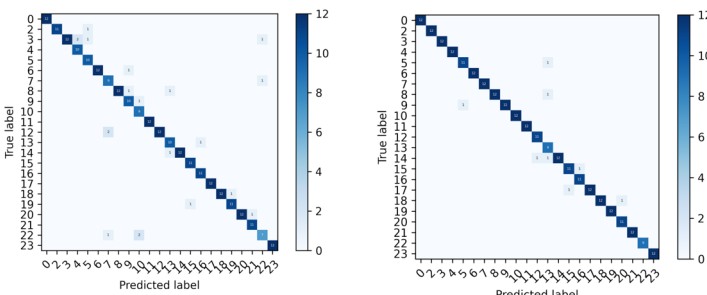

Figure 5: Confusion matrix of testing of the proposed approach. The left and right represent the ResNet-50 with triplet loss, ResNet-50 with triplet loss and ReduNet, respectively.

To evaluate the impact of triplet loss and ReduNet on classification performance, we created the confusion matrix (Figure 5) for analysis. As we can see from the figure, triplet loss and ReduNet can enhance the classifier's learning, even though the samples are imbalanced. For instance, the pure ResNet cannot model the X chromosome (labelled as 22 in Figure 5) well due to the imbalanced sample amount, and it confuses 22 with 3, 7, and 10 classes. However, with triplet loss and ReduNet, through the same ResNet network, the classifier can easily distinguish them with no error.

At last, to investigate the effect of ReduNet on representation enhancement, we applied the t-SNE (v. d. Maaten & Hinton, 2008) method to the learned representations to reduce the dimensionality for 2-D visualization. We probed the representations of testing samples learned without and with ResNet + triplet loss in Figure 6. In the figure, the left and right are the t-SNE visualizations of the learned representations of pure ResNet-50 and ResNet-50 with triplet loss + ReduNet, respectively. It illustrates that the ReduNet optimizes the structure of the representation embeddings and alleviates the entangling features as well as the presentation errors.

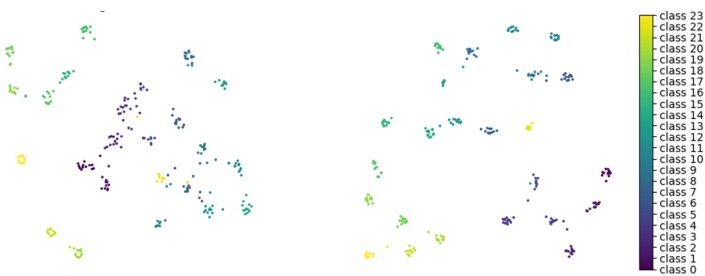

Figure 6: Representation embeddings for chromosomes using t-SNE method in test dataset. Left represents the embeddings learned by ResNet-50, the right is the learned embeddings of Resnet-50 with triplet loss + ReduNet.

## 5 CONCLUSION

This paper presents a novel three-stage pipeline for automatic chromosome classification. Unlike existing methods, the proposed approach is primarily based on learning meaningful data representations rather than only finding classification features in given samples. It integrates triplet loss and ReduNet with convolution neural networks for better feature extraction and representation. Based on the enhanced feature representation, simple and efficient machine learning algorithms are adequate for the classification task. Moreover, the proposed approach does not require manual feature engineering and pre-processing like chromosome strengthening, making it efficient and practical for real-world applications. The experimental results show that the proposed approach can achieve a very high recognition rate of 97.22% as measured by testing accuracy and 97.18% by F-measure on a small dataset.

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
