# OpenReview forum: "Learning Discriminative Representations for Chromosome Classification with Small Datasets"
_ICLR.cc/2023/Conference — Submitted to ICLR 2023_

### Official Review · Reviewer_qFAU · 2022-10-14

**Confidence:** 4
**Correctness:** 2
**Technical Novelty And Significance:** 2
**Empirical Novelty And Significance:** 2
**Recommendation:** 1

**Clarity, Quality, Novelty And Reproducibility:**

- The related work section follows at least the two following publications almost word by word (incl. Fig 1):
[1] C. Lin et al., "CIR-Net: Automatic Classification of Human Chromosome Based on Inception-ResNet Architecture," in IEEE/ACM Transactions on Computational Biology and Bioinformatics, vol. 19, no. 3, pp. 1285-1293, 1 May-June 2022, doi: 10.1109/TCBB.2020.3003445.
[2] C. Lin, G. Zhao, A. Yin, L. Guo, H. Chen and L. Zhao, "MixNet: A Better Promising Approach for Chromosome Classification based on Aggregated Residual Architecture," 2020 International Conference on Computer Vision, Image and Deep Learning (CVIDL), 2020, pp. 313-318, doi: 10.1109/CVIDL51233.2020.00-79.
Whereas the first publication is not even cited in the proposed paper. This is a severe concern regarding plagiarism.
- There is basically no related work mentioned published later than 2018, apart from Lin et al., “Cir-net: Automatic classification of human chromosome based on inception-resnet architecture” 2020. That paper alone has 22 relevant citations on the topic of chromosome classification, so I believe some more work on the background section would help to place the study in literature and discuss its particular contributions.
- There are issues with the references. At one point Sharma et al. are referred to by the author’s first name as Monika et al. in the text. However I cannot find the publication it is referencing to (Sharma, Monika and Lovekesh Vig. Application of artificial neural networks to chromosome classification. International joint conference on neural networks (IJCNN), 2018) altogether. Maybe “Automatic Chromosome Classification using Deep Attention Based Sequence Learning of Chromosome Bands” by Sharma et al. and “Application of artificial neural networks to chromosome classification” by Errington et al. have merged. Some references have first and last name mixed up, e.g. Yann LeCun et al. 2015.
- The authors emphasize that their work differs from existing methods by learning meaningful data representations, rather than only finding features needed for classification. If the downstream task of a feature extraction pipeline is classification, the most meaningful representations are the most discriminative ones for the following classification? So how else is the meaningfulness evaluated if not through the downstream task?
- The authors mention that preprocessing such as straightening is not feasible in a clinical setting (end of Sect. 2) - but is the application of two neural networks after one another more feasible in this setting? If that is really the case, the computational load and requirements for the proposed pipeline have to be addressed by experiments and put into context to resources available in a clinical setting.
- Sect. 3.2.1, diverse representation: it states that “the dimension of features for each class/cluster should be as large as possible as long as they stay uncorrelated from other classes”. What is the argument behind wanting feature dimensions as large as possible?
- I have problems with the results data shown in Fig. 5. If the true label is on the y-axis and the predicted label is on the x-axis, I should be able to add the rows to get the total number of samples in each class. In that case the results of the two plots showing the confusion matrices of ResNet+TripletLoss and ResNet+TripletLoss+ReduNet stem from experiments performed on different test sets as the number of elements in each class varies greatly. It is more likely that the axes labels are wrong, and the summation should be done column-wise, however that still leaves only 11 samples in class 2 in the first plot, and 12 samples in class 2 in the second plot.
- Furthermore I am confused as the authors describe 22 autosome classes, labeled 0 to 21, but the plots in Fig. 5 omit class 1.
-The test set is said to be split 90:10, however given Fig. 5, there seem to be 12 samples in most classes of the test set, although there are 130 samples for most classes. I count 273 samples in the test set given this figure, which does not add up to 299 which correspond to 10% of the dataset. A validation set is not mentioned in the text.
-Additionally, the numbers of misclassified images in this plot do not add up with the numbers in Table 1. The figure caption says that it shows data of  ResNet+TripleLoss vs ResNet+TripleLoss+ReduNet, but given that the test set contains 273 or 299 samples, a performance gain in accuracy of 0.54% (Table 1. ResNet+TripletLoss vs ResNet+TripletLoss+ReduNet) corresponds to 1-2 images difference in correct classification, while this plot shows 20 vs 8 misclassified samples (~ 7% and ~3% misclassified samples), so likely this left plot actually shows either ResNet results, or ResNet+ReduNet.
- In general, I would suggest stratified sampling to split the data into train and test set, as there is a different number of samples per class, . Replication of the samples in class 23 as described in Sect. 4.2 may make sense for the training set, not for the test set.
- In Fig. 6, the authors present t-SNE visualizations of the learned representations of  ResNet vs. ResNet+TripleLoss+ReduNet and conclude that an optimized structure of the representation embeddings can be observed. This claim cannot be supported by a t-SNE visualization in general, nor can any particular pattern be observed in this particular visualization.
- Given that the test set contains 273 or 299 samples, a performance gain in accuracy of 0.54% (Table 1. ResNet+TripletLoss vs ResNet+TripletLoss+ReduNet) corresponds to 1-2 images difference in correct classification. It is hard to draw any conclusions from such a result, in particular as these results seem not to be averaged over multiple runs.

I also have some issues with the mathematical notation in Sect. 3.2.2:
- Z is a high-dimensional representation if I understand correctly.  What is \Delta Z then, used in Eqn (8)?
- I fail to see how the gradient of difference of Eqn(5) and Eqn(6) results in Eqn. (9)-(11) - what are the alphas introduced in this calculation? The relationship of these variables and constants need to be defined.
- In paragraph before Eqn. (4), what are dimensions D and d?
- In 3.2.2. \Lambda is first indexed by subscript, then it switches to superscript.


Clarity:
Mathematical notation can be clarified by minor changes, but Sect. 3.2.2. should be revisited to either possibly give the reader a more high-level overview and extend the mathematical formulations in the appendix instead of the main paper, in particular as it follows Chan et al. without introducing novel methodology.

Quality:
Quality can be improved by a more thorough literature search on related work and comparison to state of the art in chromosome classification; application to another dataset and improved presentation of the results.

Novelty:
The paper does not introduce novel methodology and given issues with the dataset size, the fact that only one dataset is used, no strong conclusions can be drawn from the proposed pipeline for the particular application.

Reproducibility:
Not enough details are reported to reproduce the pipeline fully, e.g. no further information is given on the implementation of the SVM apart from the fact that a Gaussian kernel is used.


Minor Comments:
- “Insufficient labelled” -> “insufficiently labelled” at multiple places
- “datasets” -> “dataset” in Sect. 1 before listing the contributions
- some words are spelled in American English (e.g. generalizability), others in British English (e.g. summarised), only use one kind
- Fig. 2 caption, “Learning” -> “learning”
- Fig. 1 and in particular Fig. 5 should be larger, and Fig. 3 vectorized to improve its quality
- “exemplars” -> “examples”, Sect. 3.1
-”isproduced” -> “is produced” in Sect. 3.3
In the conclusions, the authors point out that their approach is superior over others as it does not require “chromosome strengthening”, should this be “straightening”?




**Details Of Ethics Concerns:**

The related work section follows at least the two following publications almost word by word, including Fig.1 showing an example of the dataset. Note that the related work section was taken over to the extend that those two works as well as the submission even refer to Sharma et al. in the same wrong way as “Monika et al.”
Publication [2] is not even cited in the proposed paper.


[1] C. Lin et al., "CIR-Net: Automatic Classification of Human Chromosome Based on Inception-ResNet Architecture," in IEEE/ACM Transactions on Computational Biology and Bioinformatics, vol. 19, no. 3, pp. 1285-1293, 1 May-June 2022, doi: 10.1109/TCBB.2020.3003445.

[2] C. Lin, G. Zhao, A. Yin, L. Guo, H. Chen and L. Zhao, "MixNet: A Better Promising Approach for Chromosome Classification based on Aggregated Residual Architecture," 2020 International Conference on Computer Vision, Image and Deep Learning (CVIDL), 2020, pp. 313-318, doi: 10.1109/CVIDL51233.2020.00-79.




**Strength And Weaknesses:**

Exploiting small datasets is a very relevant task in biomedical applications, where annotations and labels are scarce and expensive. However this paper does not address how to use the little available data to the best extent, leaving a very small test set for a fully-supervised classification model, from which not many conclusions can be drawn with the expectation to general well to other datasets. An outdated background section makes it hard to place the proposed method within literature in this particular application and unclearities with the results make it even harder to draw conclusions from the performed experiments.


**Summary Of The Paper:**

The authors propose a fully supervised classification pipeline consisting of contrastive learning with a triplet loss using ResNet, followed by ReduNet to modify the learnt features such that a nonlinear support vector machine can be used for classification of the features. The method is evaluated on one publicly available dataset of chromosome classification (24 classes).


**Summary Of The Review:**

I have a few major issues with this submission: (1) the background section is basically taken over from a publication in 2020, so it is hard to know where this particular work is located in the current state of the art of chromosome classification; (2) the one dataset used for evaluation is very small, hence accuracy gains of 2% correspond to about 5 images, so it is hard to draw conclusions from the experiment and it cannot be expected that these results generalize to any other dataset, even more so, if they are not performed on multiple runs; (3) there are flaws in the presentation of the numerical results, the basis of the study.

---

### Official Review · Reviewer_cfhg · 2022-10-24

**Confidence:** 3
**Correctness:** 3
**Technical Novelty And Significance:** 1
**Empirical Novelty And Significance:** 2
**Recommendation:** 1

**Clarity, Quality, Novelty And Reproducibility:**

The current writing of the paper allows me to conceptually understand the method and experiments, except for the following:
- In Sec 4.2, was the data for the Y chromosome replicated before or after the 90:10 split?
- How does the training proceed? I assume it goes in three stages, corresponding to the three stages of the method.
- Does the training of the ResNet50 start from scratch (random init) or are any pre-trained weights used, if so, which ones?
- What is the batch size used?
- Is any triplet mining applied?
- What is the output dimension of the network, dimensionality of the features?
- Which variant of multi-class SVM is used? How are the hyperparameters selected (like gamma for the Gaussian kernel)? Why not a linear SVM?

With the above in mind, I would find it hard to reproduce the results.

**Strength And Weaknesses:**

Pros:
- application to an important read-world challenge
- the method does not require the chromosomes to be straightened (less preprocessing)

Cons:
- the text is not clear enough (see below) and repetitive with the setting and method exposed multiple times
- the paper explains prior art that I would expect the readers to be familiar with (triplet loss, accuracy/recall/precision/f-score), but also explains ReduNet, which I did not know and even wrongly assumed it was proposed in the paper
- weak baselines (AlexNet, VGG-16), missing baselines (PCA instead of ReduNet)
- the performance improvement does not seem very convincing to me since it is based on just several samples (13 for each autosome, 9.6 for each sexual chromosome given the 90:10 split) and the difference is small (~1.2pp over Lin et al and ~0.6pp over using a triplet loss + svm)

**Summary Of The Paper:**

The paper proposes a method to classify chromosome images consisting of three steps. First, a convolutional network (ResNet50) is trained with a triplet loss, second, the output chromosome representation is optimized via a ReduNet [Chan et al, 2021], and finally fed into an SVM with a Gaussian kernel. Both the first and second steps contribute to a representation that provides well-separated classes. The method does not require any image preprocessing. The experiments evaluate the method on a publicly available dataset.

**Summary Of The Review:**

I would not recommend accepting the paper in its current form.

The proposed method seems to me too complicated given the small improvement of the results. I would find it useful if the experimental section went deeper into checking each part of the method against better and/or stronger baselines (PCA instead of ReduNet, comparison to ResNet50 trained with triplet loss with nearest neighbor using cosine distance, use of a more recent image feature extractor, such as EfficientNet or Swin). Also I would enjoy if the writing was less repetitive and clearer regarding what is new.

Minor improvements:
- "for deferent classes" -> "for different classes"
- "isproduced" -> "is produced"
- "we uniform the input image as 224x224" -> "we resize the input image to 224x224"

---

### Official Review · Reviewer_4brb · 2022-11-03

**Confidence:** 4
**Clarity, Quality, Novelty And Reproducibility:** 1. The biggest weakness of the propos…
**Correctness:** 2
**Technical Novelty And Significance:** 1
**Empirical Novelty And Significance:** 2
**Recommendation:** 3

**Strength And Weaknesses:**

Strengths:
1. The proposed approach combines three components into an integrated method, and further comparison experiments demonstrate its superior performance compared with some other methods in chromosome classification tasks.
2. The organization is well, and it is clear to readers.

Weaknesses:
1. The biggest weakness behind this paper is the lack of novelty. The main components of the proposed method are common and existing.
2. The comparison experiments are not solid. The baseline methods are limited.

**Summary Of The Paper:**

This manuscript wants to address chromosome classification with small datasets. Unlike some automatic chromosome classification methods, this approach is primarily based on learning discriminative future representations rather than only finding classification features in given data samples. The proposed method could be summarized as divided into three steps: the first step is to extract discriminative and meaningful features of chromosomes by utilizing ResNet with triplet loss. The second step optimizes features from the first step to obtain a linear discriminative representation via maximal coding rate reduction, which ensures the clusters representing different chromosome
types are far away from each other while the same is close to each other in the cluster. The final step is to employ a classification method (such as SVM) to identify chromosomes. Then, some comparison experiments are conducted to evaluate the effectiveness of the proposed method.
I think its contributions could be summarized as follows:
1. The method proposes a representation learning-oriented approach, which forces the network to learn discriminative features rather than classification features in the small dataset.
2. This paper combines some existing methods into an integrated chromosome classification method.

**Summary Of The Review:**

The paper proposes a pipeline for the automatic classification of chromosomes, which integrates three components to learn discriminative representations. However, these components exist and are common in other tasks. These incremental improvements do not convince me to accept this paper.

---

### Decision · Program_Chairs · 2023-01-20

**Decision:**

Reject

**Justification For Why Not Higher Score:**

The paper has limited novelty and all reviewers recommended rejection.

**Justification For Why Not Lower Score:**

NA

**Metareview: Summary, Strengths And Weaknesses:**

This paper presents a method to learn discriminative representation for chromosome classification. The main weakness of the paper includes:
1) limited novelty;
2) weak experimental validation
3) concern regarding plagiarism

**Summary Of Ac-Reviewer Meeting:**

NA